# Strategies for Solubility and Bioavailability Enhancement and Toxicity Reduction of Norcantharidin

**DOI:** 10.3390/molecules27227740

**Published:** 2022-11-10

**Authors:** Qian Liu, Henglai Sun, Xinyu Li, Huagang Sheng, Liqiao Zhu

**Affiliations:** College of Pharmacy, Shandong University of Traditional Chinese Medicine, Jinan 250355, China

**Keywords:** norcantharidin, solubility, bioavailability, antitumor, structural modification, nanopreparation

## Abstract

Cantharidin (CTD) is the main active ingredient isolated from Mylabris, and norcantharidin (NCTD) is a demethylated derivative of CTD, which has similar antitumor activity to CTD and lower toxicity than CTD. However, the clinical use of NCTD is limited due to its poor solubility, low bioavailability, and toxic effects on normal cells. To overcome these shortcomings, researchers have explored a number of strategies, such as chemical structural modifications, microsphere dispersion systems, and nanodrug delivery systems. This review summarizes the structure–activity relationship of NCTD and novel strategies to improve the solubility and bioavailability of NCTD as well as reduce the toxicity. This review can provide evidence for further research of NCTD.

## 1. Introduction

Mylabris (known as ‘Banmao’ in Chinese, Figure 1) is the dry body of *Mellabris Phalerata* Pallas or *Mylabris Cichorii* Linnaeus, an insect of the meloidae family, which wasrecorded in the 2020 edition of the Chinese Pharmacopoeia. It has the effect of reducing swelling and dispersing knots, counteracting toxic substances and phagedenism. It can be used for abdominal mass, amenorrhea, stubborn dermatitis, crewels, and so on [1]. The main bioactive component of Mylabris is cantharidin (CTD) (Figure 1) [2]. Numerous studies have shown that CTD can be widely used in the treatment of many cancers [3,4,5,6,7], but it can cause severe cardiotoxicity [8], hepatotoxicity [9], and nephrotoxicity [10]. Therefore, its clinical application is limited. In recent years, norcantharidin (NCTD) (Figure 1), a demethylated derivative of CTD, has become a research hotspot because of its wide range of biological activities, such as treating systemic lupus erythematosus [11], antiplatelet aggregation [12], antirenal interstitial fibrosis [13,14,15], and antitumor activity [16,17].

NCTD can reduce hepatotoxicity and nephrotoxicity compared with CTD [18,19]. In the meantime, NCTD can still inhibit the growth of various cancer cells or induce apoptosis, such as prostate cancer [20], cholangiocarcinoma [17], colon cancer [21], breast cancer [22], neuroblastoma [23], osteosarcoma [24], etc. Despite its extensive antitumor activity, NCTD still has toxicity like nephrotoxicity [25], hepatotoxicity [26], urological toxicity, and cardiotoxicity [27], and it can be an irritant to the injection site [28]. In addition, NCTD has a short elimination half-life (intravenous administration to Sprague-Dawley (SD) rats, t_1/2(β)_ = 2.63 h) [28] and poor solubility (2.5 mg/mL at pH 6.0) [29]. To improve the safety and efficacy of NCTD treatment, scholars have conducted extensive experimental studies. Among various strategies, structural modifications and nanodrug delivery systems have made remarkable achievements in improving the solubility and bioavailability of NCTD. This review summarizes the strategies for NCTD to improve its solubility and bioavailability and reduce the toxicity (Figure 2).
Figure 1The origin and chemical structure of NCTD.
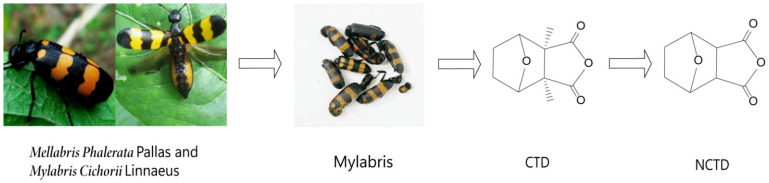



## 2. The Structure–Activity Relationship of NCTD

NCTD is composed of an oxygenated six-membered ring and dicarboxylic anhydride. The bridging oxygen and the anhydride part in its structure play an important role in reducing toxicity and improving the antitumor activity. Structure–activity relationship studies showed that the removal of bridging oxygen on the ring could cause a reduction or disappearance of cytotoxicity. The presence of the dicarboxylic anhydride structure could retain or enhance the antitumor activity of NCTD [30]. Yeh et al. [31] found that the removal of bridging oxygen on the ring, retaining the anhydride structure, could not only reduce hepatocellular toxicity to primary rats, but also significantly induce apoptosis in liver cancer cells. Moreover, the ether oxygen atom on the anhydride structure can be replaced by the nitrogen-containing atomic group to obtain NCTD derivatives, and the anhydride opens the ring to form acid amide or dicarboxylic acid derivatives of NCTD, which is conducive to enhance antitumor effect. In the meantime, the C5/C6 position of NCTD is also a key modification site to enhance antitumor activity. The structure–activity relationship and modification strategies are shown in Figure 3.

## 3. Chemical Structure Changes

The chemical structure change of NCTD is an important strategy to improve its solubility and membrane permeability, which is beneficial to improve the bioavailability of NCTD and enhance the bioactivity [32]. The main modification sites of NCTD structure are the dicarboxylate anhydride structure and the C5/C6 position (Figure 3). The structural modification strategies of NCTD include derivatives and prodrug strategies, both of which can effectively improve the bioavailability of NCTD and reduce its toxicity.

### 3.1. Derivatives

Derivatives are advantageous ways by modifying the structure of drug parents, changing drug production performance to improve bioavailability, reduce toxicity, and enhance therapeutic effects [33,34]. At present, NCTD derivatives were widely studied. It was reported that the structural modification of NCTD could obtain more excellent and bioeffective derivatives of NCTD. The preparation methods of NCTD derivatives include structural modification of dicarboxylate anhydride and modification at the C5/C6 position.

#### 3.1.1. Structural Modification of Dicarboxylate Anhydride

##### Cyclolactone Derivatives of NCTD

Fostriecin (Figure 1 and Figure 4) is a topoisomerase II catalytic inhibitor with strong antitumor effect that has entered phase I clinical trials [35]. Structure–activity relationship studies showed that the cyclolactone structure of fostriecin is advantageous for improving the antitumor potency [36]. To better simulate the antitumor activity of fostriecin, Tarleton et al. [37] synthesized compound (3*S*,3a*R*,4S,7*R*,7a*S*)-3-hydroxyhexahydro-4,7-epoxyisobenzofuran-1(3*H*)-one(2), which had a cyclolactone structure similar to fostriecin. Compared with NCTD, compound **2** (Figure 4) showed similar antitumor activity to nine cancer cell lines. To obtain derivatives with stronger antitumor activity, compound **2** was further structure-modified. The results indicated that the isopropyl-substituted compound **3** (Figure 4) showed strong cytotoxicity to colon cancer cell line HT29 and glioblastoma cell line SJ-G2. In addition, compound **4** (Figure 4) with the terminal phosphate group displayed the greatest growth inhibition to neuronal carcinoma cell line BE2-C (GI50 = 9 μM).

##### Norcantharimide Derivatives

Studies have proved that NCTD could react with a range of amines to obtain the corresponding norcantharimides (Figure 4) to improve the bioactivity, and these norcantharimides had moderate to high cytotoxicity to a variety of cancer cells. Among them, derivatives of norcantharimide with C10 (Figure 4 and Figure 5), C12 (Figure 4 and Figure 6), C14 alkyl chain (Figure 4 and Figure 7), or a dodecyl-linked second norcantharimide moiety (Figure 4 and Figure 8) showed the highest level of cytotoxicity. However, the substitution of the shorter alkyl chain the position of the nitrogen atom of NCTD was detrimental for enhancing antitumor activity [32]. Robertson et al. [38] synthesized compound **9a** (Figure 4) with a terminal phosphate group by using an amino-group substitution and a phosphorylation reaction, which showed stronger antitumor effect to nine cancer cell lines compared with NCTD. In addition, compounds **10(a–e)** (Figure 4) with the diphenyl phosphate group and compounds **11(a–e)** (Figure 4) with the trichloroethyl phosphate group showed strong antitumor activity to almost all cancer cell lines. Among them, the compounds **10b**, **10d**, and **10e** with long fatty chains displayed 2–5 times increased antitumor potency compared with NCTD. These indicated that compounds with diphenyl phosphate groups and trichloroethyl phosphate groups were more hydrolyzed relative to simple alkyl esters, exposing free phosphate, and thus exhibiting significant antitumor activity. The lipophilicity of *N*-substituted norcantharimide derivatives play a crucial role in their bioactivity. Wu et al. [39] prepared compound **12** with the terpene group and compound **13** with the terpene oxygen group (Figure 4), which were both 2–5 times more cytotoxic than NCTD. At the highest concentration of 60 μM, compounds **12** and **13** showed no toxicity in the normal mouse embryonic liver BNCL.2 cell line. Pachuta-Stec et al. [40] prepared compound **14** containing an *N*-substituted amide group and 1,2,4-triazole ring by condensation and nucleophilic reactions (Figure 4), which had high cytotoxic and antiproliferative effects on the human hepatocellular cancer cell line Hep3B, but did not affect the activity of normal human liver stellate cell line LX-2. Therefore, it is a promising antitumor drug.

Chang et al. [41,42,43] synthesized *N*-farnesyloxynorcantharimide (NOC15) and *N*-farnesyl-norcantharimide (NC15) and investigated their antitumor effects. The results showed that NOC15 and NC15 had better safety and efficacy compared with NCTD when treating human acute leukemia Jurkat T cells. The antitumor activity of NOC15 was 1.68 times higher than that of NC15. Antitumor mechanism studies revealed that NOC15 inhibited the growth of cancer cells by increasing the percentage of subG1 phase cells, activating p38 and ERK1/2 signaling pathways, and inhibiting CaN expression and IL-2 production.

Structure–activity relationship studies indicated that the retention of sp2 hybrid junctions in the tetrahydroepoxy isocolenol carboxylamide scaffold of norcantharimide derivative was crucial, and that the reduction of aromatization of phenolic functions and the removal of single olefin bonds could significantly reduce the antitumor effect. In relation to stereochemistry, the trans isomers typically showed marginally increased cytotoxicity, whilst the attachment of additional hydrophobic moieties to the oxabicyclo region resulted in a slight decrease in potency (Figure 5). Spare et al. [44] believed that adding hydrophobic groups (alkanes, olefins, and aromatic groups) within the tetrahydroepoxy isocolenol carboxylamide scaffold of norcantharimide derivatives is beneficial to improving the antitumor activity. Among these, the compound **15** (Figure 4) displayed the highest cytotoxicity to breast cancer cell line MCF-7 and HT29 cells, which had GI50 values of 6.5 μM and 11.3 μM, respectively. However, norcantharimide derivatives containing hydroxyl and dimethyl amino groups displayed less tumor-suppressive effects.
Figure 4Cyclolactone and norcantharimide derivatives of NCTD.
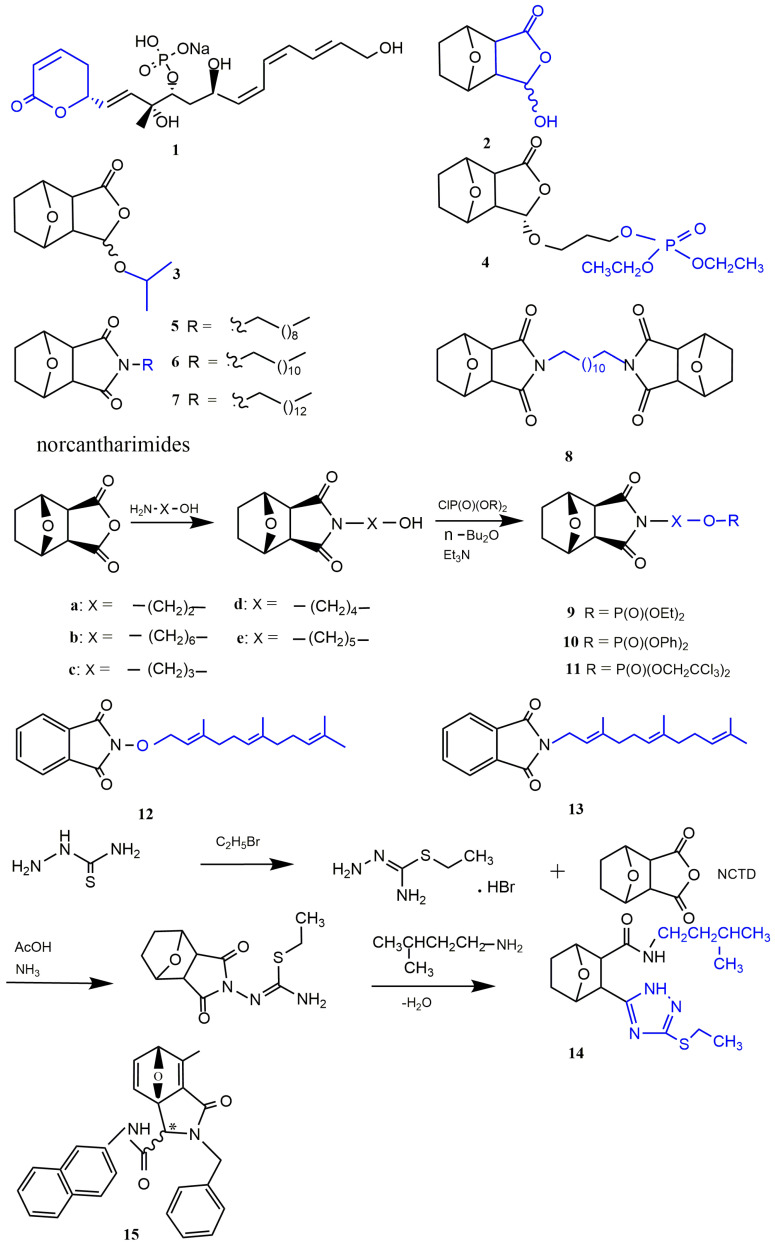

Figure 5Structure–activity relationship of norcantharimide derivative with the tetrahydroepoxy isocolenol carboxylamide scaffold.
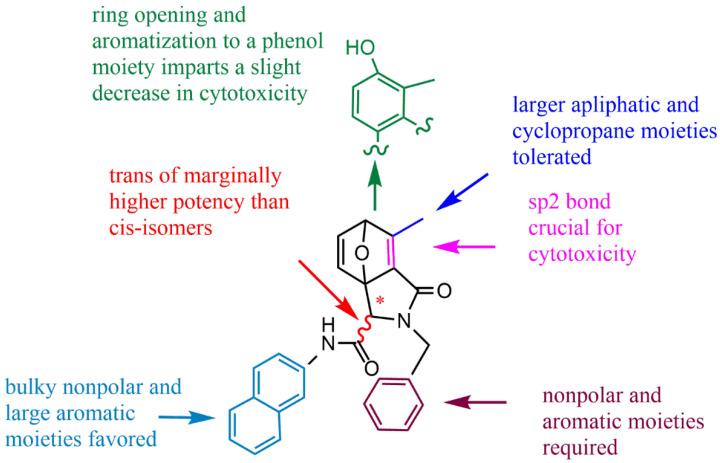



##### Acid Amide Derivatives of NCTD

Open-loop acid amide derivatives of NCTD containing a benzenesulfonamide group were obtained by the reaction of NCTD with substituted benzenesulfonamides (Figure 1 and Figure 6). Compared with NCTD, compound **1** not only improved solubility, but also enhanced cytotoxicity to a variety of cancer cells and reduced toxicity to human lung fibroblast cell line WI-38. Furthermore, compound **1** could significantly inhibit protein phosphatase 1 (PP1) activity and microtubule formation. Therefore, it could be used as a potential PP1 inhibitor in antitumor drugs [45]. To further investigate antitumor effects of NCTD derivatives, Hizartzidis et al. [46] synthesized a series of acid amide derivatives of NCTD, among which only the compound **2** (Figure 6) showed high cytotoxicity to some cancer cell lines. To explore the effect of furan fractional additional substitution and biphenyl substitution on antitumor activity, the compounds with a furan structure were synthesized. It was obvious that only the compound **3** (Figure 6) with the biphenyl moiety showed significant growth inhibition to HT29 cells, glioblastoma cancer cell line U87 and ovarian cancer cell line A431. Among the synthesized compounds with pyrrole and biphenyl structures, compound **4** (Figure 6) had stronger antitumor activity than compound **5** (Figure 6) with an additional benzene structure [47]. Therefore, these results indicated that the biphenyl substitution was advantageous for enhancing the cytotoxicity of acid amide derivatives of NCTD, and the introduction of additional furan groups or the benzene moiety was detrimental to enhancing the antitumor activity.
Figure 6Acid amide derivatives of NCTD.
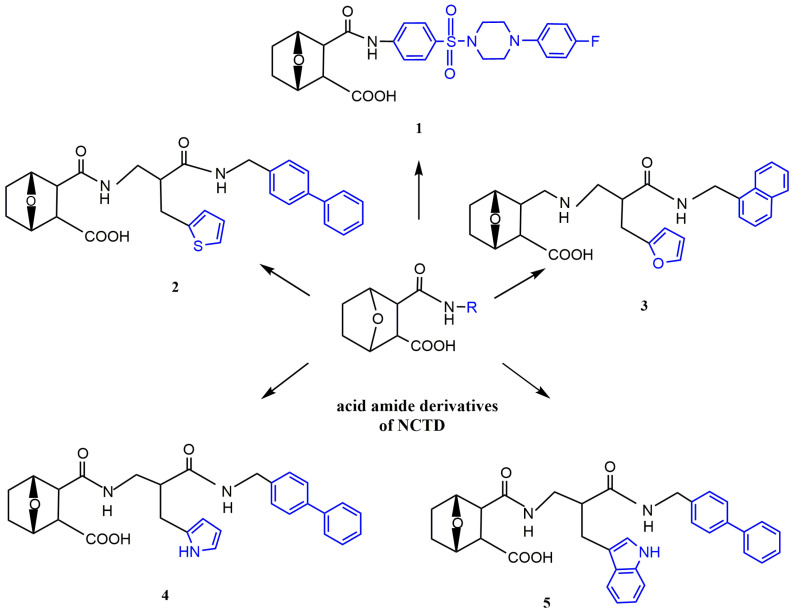



##### Dicarboxylic Acid Derivative of NCTD

NCTD could couple with camptothecin (CPT) to generate dicarboxylic acid derivatives of NCTD to enhance the antitumor effect. The investigators synthesized a series of CPT-HAA-NCTD-conjugated compounds **1(a–f)** (Figure 7). This series of compounds showed high solubility and strong activity against several cancer cell lines [48]. Zhao et al. [49] prepared compounds **2(a–g)** (Figure 7), which not only showed strong inhibition to several cancer cell lines, but also displayed synergistic effects. In addition, Zhao et al. [50] prepared nine conjugates of camptothecin and NCTD linked by alanine (CPT-Ala-Nor) (Figure 7), which displayed significant inhibition to human hepatoma cell line HepG2, human colon carcinoma cell line SW480, human gastric carcinoma cell line BGC803, and human pancreatic cell line PANC-1. Structure–activity relationship studies indicated that compounds **3(d–g)** (Figure 7) had higher activity against HepG2, BGC803, SW480, and PANC-1 cells than compounds **3(a–c),** which was related to the disappearance of the double bond at the C5/C6 position of NCTD in compounds **3(d–g)**. The order of antitumor activity of these conjugates followed as 5,6-dibrom (compound **3g**) > 5,6-2H (compounds **3d–3f**) > 5,6-ene (compounds **3a–3c**). Similarly, compounds **4(a–f)** and **5** (Figure 7) also had strong antitumor effects [51,52]. Dong et al. [53] prepared conjugates **6(a–c)** (Figure 7) of NCTD and icaritin, which could significantly inhibit HepG2 and MCF-7 cells.

In conclusion, the conjugates of NCTD and small-molecule drugs had significant antitumor potential and might serve as a promising antitumor drug. However, the structure–activity relationship of these conjugates still needs further investigation.
Figure 7Dicarboxylic acid derivatives of NCTD.
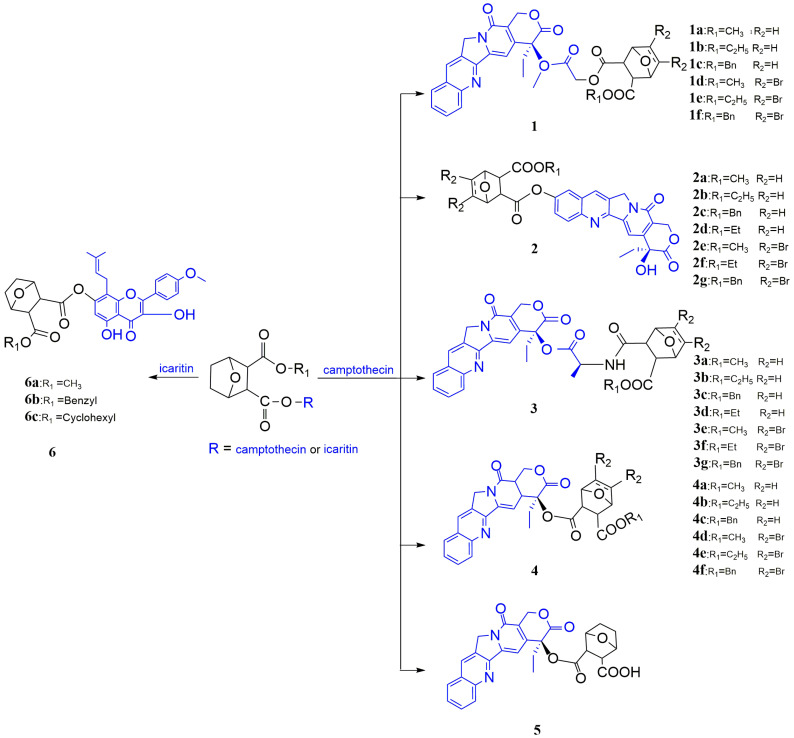



#### 3.1.2. Modification at the C5/C6 position

The C5/C6 position of NCTD is a key site for structural modification, and it can be replaced by some pharmacophore groups, such as imidazolium salts and pyrazole and isoxazole groups, to improve the antitumor activity of NCTD. Studies have shown that imidazole salts were an important component in the structure of chemotherapeutic agents and played an important role in antitumor activity [54,55]. Sun et al. [56] prepared *N*-substituted imidazole NCTD derivatives to investigate their antitumor potential. The results demonstrated that spatial and electron effects play a key role in exerting antitumor activity of imidazolium salts. Compounds **1(a–f)** and **2(f–h)** (Figure 8) had the electron-donating group, carbonyl, and propenyl, which displayed high cytotoxicity to human epidermal carcinoma (Hela). Therefore, *N*-substituted imidazole NCTD derivatives were found to be the most potent antitumor compounds. To explore higher antitumor activity of NCTD derivatives, Wang et al. [57,58] synthesized a series of NCTD derivatives containing chromone linked pyrazole or linked isoxazole moiety, compounds **3–8(a–f)** (Figure 8), but their bioactivity needs further experimental studies.
Figure 8NCTD derivatives modified at the C5/C6 position and their synthetic pathway.
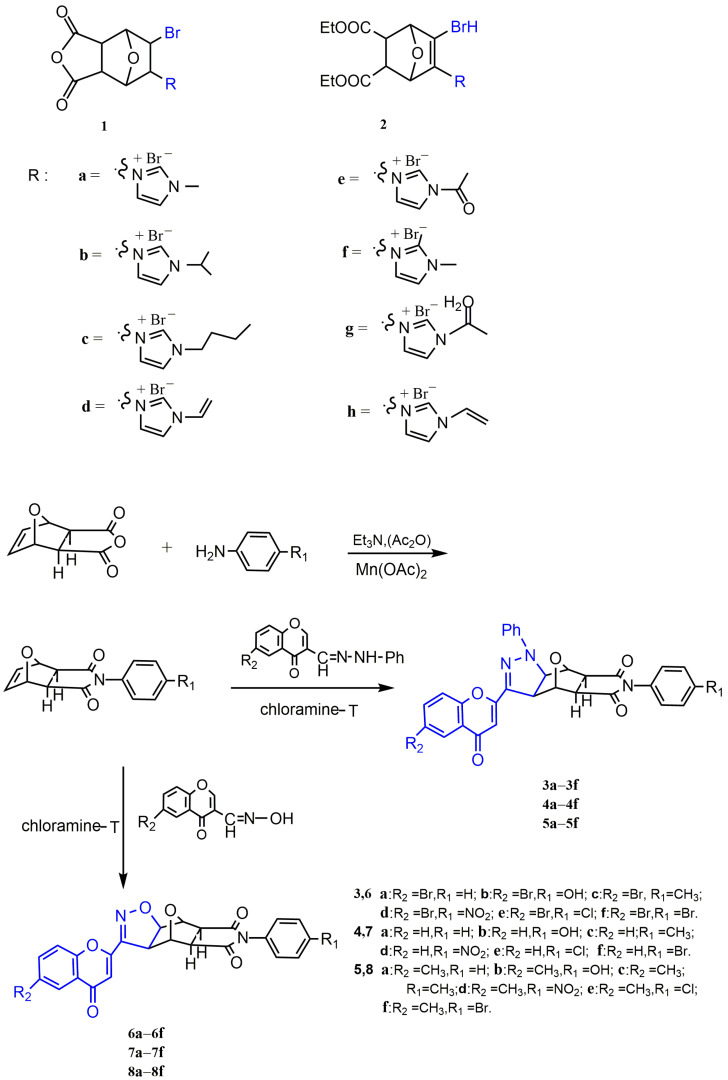



### 3.2. Prodrug Strategies

Structural modifications also use biosynthesis by adding precursors to small-molecule drugs. Studies have proved that NCTD can form prodrugs by combining with chitosan derivatives or podophyllotoxin, thus solving the problems of poor solubility, low bioavailability, and high toxicity of NCTD.

#### 3.2.1. The Conjugates of NCTD with Chitosan and Its Derivatives

Chitosan (CS) is a biodegradable polymer with excellent performance, which combines with small-molecule drugs to produce useful prodrugs, showing strong bioactivity at the target site, which can increase the accumulation of drugs in tumor tissues, so as to improve the efficiency of drug delivery and reduce the toxicity of drugs to normal tissues [59]. Xu et al. [60] combined NCTD with chitosan (NCTD-CS) to obtain a better water-soluble prodrug, which could solve the problem of poor solubility and promote NCTD delivery in vivo. Moreover, NCTD-CS was precisely localized to the lysosomes in human gastric cancer cell line MGC80-3 through endocytosis, which was beneficial to increase drug accumulation and improve the efficacy. Li et al. [25] obtained a prodrug **1** (NCTD-CS) by the alcohololysis reaction (Figure 9), which prolonged the circulation time by a sustained releasing NCTD from NCTD-CS, thus reducing the toxicity of NCTD. In addition, in vitro and in vivo studies have shown that prodrug **1** had stronger tumor inhibition compared with free NCTD.

Hydroxypropyltrimethyl ammonium chloride chitosan (HACC) is a derivative of chitosan, which has strong electrostatic interactions with negatively charged tumor cells, thus enhancing the drug–polymer cytotoxicity to cancer cells. Therefore, it can be used as a formulation carrier and a delivery system for therapeutic molecules. Xu et al. [61] prepared a water-soluble prodrug **2** (NCTD-HACC) (Figure 9), which released NCTD through a biphasic drug release mode, improving the selective accumulation of NCTD at the tumor site and the therapeutic efficacy.

Carboxymethyl chitosan (CMCS) is a water-soluble derivative of CS, with better biological properties compared with CS [62]. Jiang et al. [63] combined NCTD with CMCS to obtain prodrug **3** (NCTD-CMCS) (Figure 9), which had stronger antitumor activity and lower hepatotoxicity compared with free NCTD. Chi et al. [64] demonstrated that prodrug **3** (Figure 9) could significantly inhibit the growth and metastasis of tumor cells both in vitro and in vivo and prolong the survival of tumor-bearing mice. These results indicate that prodrug **3** may be a promising antitumor drug. In addition, Chi et al. [65] found a new prodrug (CNC) of NCTD and CMCS conjugating (Figure 4 and Figure 9), which showed significant anti-proliferative activity to human gastric cancer cell line SGC-7901. CNC also had stronger antitumor effects than NCTD in BALB/c nude mice carrying SGC-7901 cells, and no weight loss was observed in the CNC-treated group of nude mice. In the study of treatment of liver cancer, CNC showed significant anti-proliferative activity and the ability to induce apoptosis to human hepatocellular carcinoma cell line BEL-7402. The inhibitory effect of CNC on tumor tissue growth in H22 hepatocellular carcinoma model was stronger than that of free NCTD. The results of the pharmacokinetic and tissue distribution analysis showed that CNC could increase the effective accumulation of NCTD in the tumor region, prolong its retention time and reduce cardiotoxicity and nephrotoxicity [27].

In brief, the conjugates of NCTD with chitosan and its derivatives can improve NCTD biodistribution, reduce the toxicity of NCTD, and prolong blood circulation time, thus enhancing antitumor activity.

#### 3.2.2. The Conjugates of NCTD with Phostophyllotoxin

Another structural modification is the combination of NCTD with small-molecule antitumor drugs into prodrugs to reduce its toxicity and improve antitumor activity. Han et al. [66] synthesized 18 podophyllotoxin–norcantharidin hybrid prodrugs. Among them, prodrug **5** (Figure 9) could cause cell cycle arrest and induce apoptosis of MCF-7 cells by upregulating CDK1 and downregulating CyclinB1 expression. Meanwhile, it exhibited higher antitumor activity and lower toxicity compared with free NCTD. Tang et al. [67] demonstrated that prodrug **6** (Figure 9) had stronger inhibition to topoisomerase Ⅱ and protein phosphatase 2A.
Figure 9Prodrugs of NCTD.
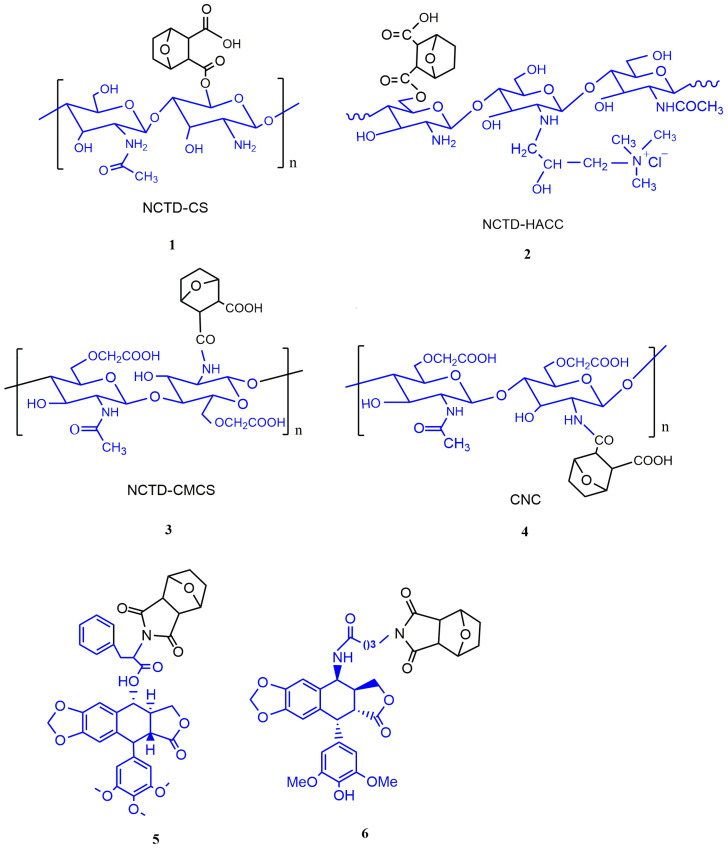



## 4. Microsphere Dispersion Systems

Microspheres refer to a particle dispersion system that buries drugs into a macromolecule or polymer matrix with a diameter of 1–1000 μm. Microspheres can solve the problems of low drug-targeting efficiency, low bioavailability, and high toxicity. Liu et al. [68] prepared composite PLGA-alginate microspheres loaded with NCTD, which could inhibit tumor growth and prolong mouse survival in a transplanted rat model of liver cancer compared with free NCTD. Previous studies have indicated that biodegradable polymeric microspheres could control the rate of drug release and prolong the biological half-life, thereby reducing the frequency of administration and the toxicity of drugs [69,70,71]. Yan et al. [72] prepared NCTD albumin microspheres with better biocompatibility, which could significantly improve liver tumor targeting of the drugs, rapidly distribute NCTD in the body, and prolong blood circulation time of NCTD compared with free NCTD. Zhang et al. [73] evaluated the effect of loading NCTD sustained-release microspheres of alginate-chitosan (NCTD/LSD-ACMs) on transarterial hepatic chemoembolization in a VX2 rabbit liver cancer model. The results indicated that NCTD/LSD-ACMs showed significant inhibition to tumors after embolization, while the inhibitory effect of NCTD on tumors was not obvious. Wang et al. [74,75,76] prepared Poly (ε-caprolactone) (PCL) microspheres encapsulating disodium norcantharidate (DSNC), an analogue of NCTD, which could slowly release DSNC. Therefore, DSNC-loaded PCL microspheres may be beneficial to reduce toxicity of DSNC on the gastrointestinal and urinary tracts, but they need further experiments.

In conclusion, NCTD microspheres can play an important role in reducing the toxicity of NCTD and enhancing the antitumor activity.

## 5. Nanodrug Delivery Systems

Nanodrug delivery system (NDDS), usually referring to loading the drug on a nanocarrier of 1–1000 nm in diameter and delivering them to a tissue or organ [77], is a beneficial tool to improve the stability and bioavailability of natural active ingredients. Nanocarriers are usually developed from lipids, proteins, sugars, polymers, metal, or fiber materials. Compared with traditional drug carriers, nanocarriers can overcome the pharmacological limitations of poor stability and low bioavailability [77,78,79,80]. Targeted nanoformulations can not only utilize the natural properties of nanocarriers to deliver drugs to specific sites, but also improve the therapeutic efficacy of drugs and reduce toxicity of drugs [81]. Currently, there are several NDDS of NCTD, including nanoparticles, liposomes, lipid microspheres, polymeric micelles, and thermosensitive gel.

### 5.1. Nanoparticles

In the past decade, nanoparticles have been widely used as drug delivery systems for the treatment and diagnosis of diseases [82,83,84]. The preparation and modification of nanoparticles can change drug performance and accurately localize to the lesion site, thus enhancing the efficacy and safety of the drugs [85,86,87]. Ding et al. [28] coated PVPk30 on the surface of NCTD chitosan nanoparticles (PVP-NCTD-NPs) to enhance the stability of NCTD, which increased the elimination half-life of NCTD and prolonged the body circulation time, subsequently improving the bioavailability. Wang et al. [88] encapsulated NCTD in galactosylated chitosan (GC) to obtain biocompatible NCTD-GCNPs, which could accurately deliver NCTD to liver cancer cells, effectively improve the local concentration of NCTD in the tumor, improve the intestinal absorption of NCTD, and reduce the uptake of normal cells, thus enhancing its efficacy and reducing the toxicity [89]. Furthermore, a novel NCTD liver-targeted drug delivery system (Lac-NCTD-TMC-NPs) had good solubility and liver targeting, which reduced the passive targeting of NCTD, thus reducing the toxicity and improving the inhibition of hepatoma cells [90].

Glycyrrhizoic acid (GA) can be used as a targeting ligand. Nanoparticles encapsulating chemotherapeutic drugs are modified by GA, which demonstrated higher liver targeting and antitumor activity [91,92]. Zhang et al. [93] prepared a NDDS with a glycyrrhetinic acid (GA)-decorated copolymer (mPEG-PCL-PEI-GA, MPG) to deliver NCTD. Compared with free NCTD, GA-conjugated nanoparticles loaded with NCTD could not only overcome the limitation of the poor solubility of NCTD, but also enhance liver tumor targeting. The antitumor activity studies implied that the nanoparticles of NCTD exhibited strong cytotoxicity to HepG2 cells and could inhibit tumor proliferation and prolong the survival of tumor-bearing mice. Li et al. [94] used commercial ligand-RGD motif (Arg-Gly-Asp) to modify lipid–polymer hybrid (LPH) nanoparticles (RGD-LPH) and deliver NCTD through targeted Integrin 5 (ITGA5) to treat metastatic triple-negative breast cancer. The results showed that RGD-LPH-NCTD accumulated more efficiently in nude mice and suppressed breast tumor growth and lung metastasis compared with free NCTD and LPH-NCTD.

Huang et al. [95] synthesized a novel Strontium/Chitosan/Hydroxyapatite/NCTD composite, which could promote osteogenesis and inhibit the proliferation of bone tumor cells. The action mechanism might be the induction of apoptosis through upregulating the caspase-3 expression and decreasing osteoclast activity of tumor cells by downregulation of osteoclast-related genes. Moreover, its therapeutic effect was closely related to the content of NCTD and the number of nanoparticles in the composite. Yan et al. [96] prepared a modified lipid nanoparticle (NLC) that encapsulated NCTD, which significantly inhibited HepG2 cell proliferation and induced apoptosis compared with free NCTD.

ABT-737 is inhibitor of members of the Bcl-2 family of apoptosis regulators, which can be used as a promising antitumor drug [97]. Studies have reported that NCTD combined with ABT-737 treatment had stronger antitumor effects compared with NCTD or ABT-737 alone [98,99]. Liu et al. [100] prepared a folate acid (FA)–lipid bilayer (LB)–chlorodimethyloctadecylsilane (CH)-coated mesoporous silica nanoparticle (MSN) (FA-LB-CHMSN) (Figure 10). The diacid metabolite of NCTD (DM-NCTD) was encapsulated in CHMSN, while ABT-737 was between the lipid bilayers. Compared with DM-NCTD or ABT-737, FA-LB (ABT-737)-(DM-NCTD@CHMSN) not only had sustained release and tumor-targeting ability in H22 tumor-bearing mice, but also prolonged the duration of action, enhanced antitumor activity, and reduced the toxicity. In addition, NCTD/tetrandrine (Tet) dual drug-loaded lipid nanoparticles (FA-LP@Tet/(MSNs@NCTD)) based on mesoporous silica nanoparticles (MSNs) (Figure 11) also exhibited sustained release ability, higher targeting, and lower toxicity compared with free NCTD. FA-LP@Tet/(MSNs@NCTD) showed strong tumor-suppression ability by reversing the multidrug resistance [101]. Xie et al. [102] prepared NCTD/paclitaxel (PTX)-loaded core-shell lipid nanoparticles modified with a tumor neovasculature-targeted peptide (Ala-Pro-Arg-Pro-Gly, APRPG) and investigated their antitumor effects in hepatocellular carcinoma (HCC). In vitro and in vivo studies showed that PTX/NCTD-APRPG-NPs had the significant ability of sustained release and liver targeting. Compared with PTX/NCTD-NP group, PTX/NCTD-APRPG-NP could remarkably inhibit the proliferation and migration of HCC cells.
Figure 10A schematic illustration of the synergistic co-delivery of diacid metabolite of NCTD and ABT-737 based on folate-modified lipid bilayer-coated mesoporous silica nanoparticle. Adapted from Ref. [100].
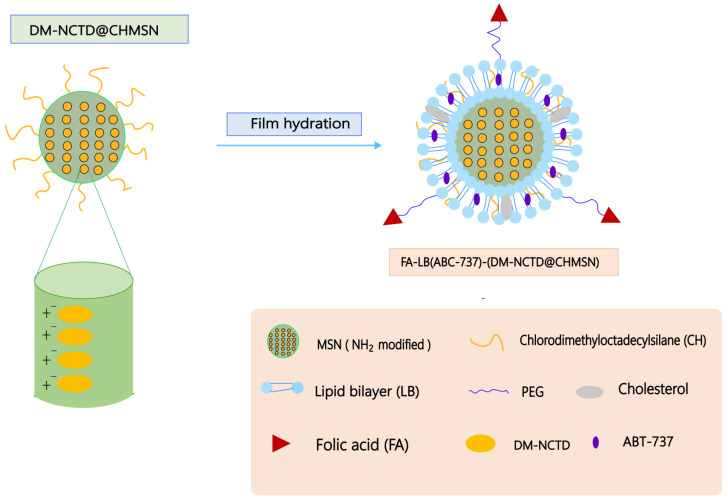

Figure 11The synthetic route of FA-LP@Tet/(MSNs@NCTD). Adapted from Ref. [101].
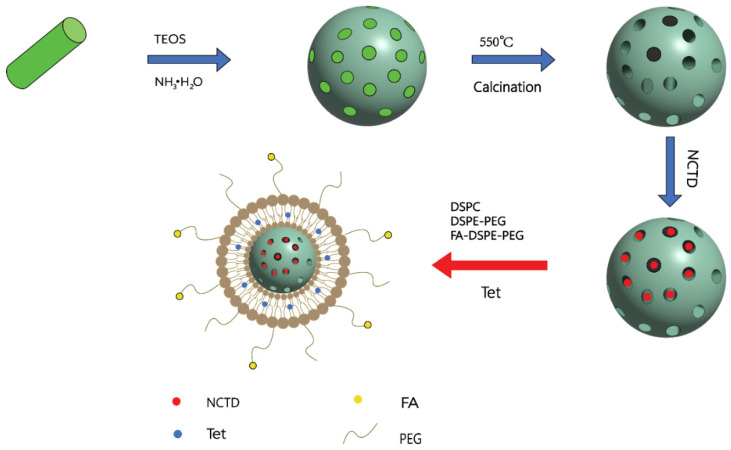



Previous studies have reported that the Zn (II)-based coordination polymer had significant antitumor activity compared with maternal drugs and could reduce toxicity [103,104,105]. Wang et al. [26] selected NCTD as an organic ligand and synthesized two metal-organic coordination polymers, DMCA-Zn1 and DMCA-Zn2, with zinc nitrate hexahydrate, which were then transformed into stable nanoparticles (DMCA-Zn1NPs and DMCA-Zn2NPs). Compared with free NCTD, DMCA-Zn1NPs and DMCA-Zn2NPs prolonged the retention of NCTD, increased the accumulation in the tumor regions, and significantly inhibited Hep3B cell proliferation. Meanwhile, it also reduced both hepatotoxicity and nephrotoxicity.

In brief, nanoparticles loaded with NCTD can not only improve the bioavailability of NCTD and reduce the toxicity, but also play an important role in the sustained release and targeted delivery of NCTD, thus enhancing antitumor efficacy.

### 5.2. Liposomes

In nanoparticle-mediated delivery systems of chemotherapeutic drugs, liposomes are the most widely used drug carriers and are one of the nano-scale drug carriers in clinical applications [106,107]. The amphiphilic, nontoxic, biodegradable, and specific targeting characteristics of liposomes promote the precise targeted delivery of drugs, thereby providing better safety in treatment. In addition, they can enhance the solubility and bioavailability of drugs and prolong the release time of drugs [108,109,110]. In order to improve the targeting of liposomes, researchers have tried to functionalize their surface to achieve the purpose of active targeting in recent years. Zhang et al. [111] prepared NCTD liposomes modified with a novel mouse anti-human CD19 monoclonal antibody 2E8 to improve specific targeting and reduce nonspecific cytotoxicity. The results showed that 2E8-NCTD-liposomes had higher cytotoxicity to human leukemia cell line Nalm-6 and lower cytotoxicity to nonspecific Molt-3 lymphoblast cells compared with the same concentration of NCTD liposomes or free NCTD groups. In the study of B-lineage leukemia stem cell line B-LSCs, 2E8-NCTD-liposomes showed the ability to specifically target B-LSCs cells and induce apoptosis. The mechanism of inducing apoptosis was related to the downregulation of HLF expression and the upregulation of NFIL3 expression [112]. Moreover, the liposomes encapsulated in NCTD were modified by the novel humanized anti-human CD19 monoclonal antibody Hm2E8b, which had the potential to specifically kill B-LSCs cells and reduce nonspecific cytotoxicity [113]. The action mechanism was related to targeting the HLF-SLUG axis. These results indicated that NCTD-liposomes modified with the CD19 monoclonal antibody not only had high targeting efficiency and specific cytotoxicity, but also could reduce the toxicity and enhance the antitumor efficacy. Zhu et al. [114] found that stearate (SG)-modified NCTD liposomes (SG-NCTD-LIP) could prolong NCTD release from liposomes. Compared with NCTD-LIP and free NCTD, SG-NCTD-LIP significantly enhanced the cytoxicity to HepG2 cells, which might be related to the interaction between SG exposed to the liposome surface and GA receptors on the cell membrane.

In order to improve the encapsulation rate of NCTD and sterilization stability, reduce the toxicity, and enhance the antitumor activity, Liu et al. [115] prepared a novel NCTD liposome–emulsion hybrid (NLEH) delivery system. NCTD was loaded in the oil-phase and oil–water interface layers as phospholipid complexes and encapsulated in a phospholipid bilayer. Compared with free NCTD, NLEH could enhance antitumor activity by promoting tumor absorption, sustained releasing drugs, prolonging blood circulation, reducing elimination, enhancing tumor targeted accumulation, improving tumor permeability, and improving antitumor immunity. In addition, the targeting of NLEH in heart and kidney was significantly reduced, thus reducing the hepatotoxicity and cardiotoxicity in vivo.

Studies have shown that PH-sensitive liposomes are extremely unstable in the acidic environment surrounding tumors, which can actively release drugs, increase the accumulation of drugs in the blood and controlling drug release, thus enhancing the therapeutic effect [116,117]. Qiao-ling et al. [118] combined synthetic lactosyl-NCTD (Lac-NCTD) with phospholipids to form liposomes containing the lactosyl-NCTD phospholipid complex (LPC) and modified the loaded liposomes through carboxylated chitosan (CMCT) to obtain pH-sensitive LPC liposomes (pH-LPC-lips). The experimental results showed that the capture efficiency of pH-LPC-lips (70.00% ± 1.30%) was significantly higher than that of Lac-lips (38.12% ± 1.12%), which might be related to the enhanced lipophilicity of LPC. Moreover, pH-LPC-lips increased the concentration by actively releasing NCTD around the tumor, thereby enhancing the cytotoxicity and tumor-suppressor effects.

The diacid metabolite of NCTD (DM-NCTD) is a stable form of NCTD. However, its clinical application was limited by its low bioavailability, short half-life, and poor safety [119]. Therefore, Liu et al. [120] encapsulated DM-NCTD in polyethylene glycol (PEG)-liposomes (DM-NCTD/PEG-liposomes) or folic acid (FA)-PEG-liposomes (DM-NCTD/PA-liposomes) to improve the bioavailability of DM-NCTD. Compared with DM-NCTD group, both DM-NCTD/PEG-liposome and DM-NCTD/PA-liposome groups could prolong the circulation time of DM-NCTD in the blood and improve the bioavailability. To solve the problems of low bioavailability and high toxicity of NCTD, Liu et al. [121] prepared asialoglycoprotein receptor-targeted, galactosylated liposomes loaded with *N*-14NCTDA to reduce the toxicity and improve the therapeutic effect of liver cancer. Compared with unmodified liposomes, the encapsulation efficiency of GAL-Lipo significantly increased to more than 98.0%. Furthermore, GAL-Lipo could prolong the half-life and improve the bioavailability, thereby achieving higher therapeutic efficiency and less hepatorenal toxicity compared with free *N*-14NCTDA. Compared with conventional liposomes (Con-LPs) and GAL-Lipo, the liposomes modified with SP94 and loaded with *N*-14NCTDA (SP94-LPs) displayed stronger cytotoxicity to HepG2 cells, which might be due to the rapid internalization of numerous SP94-LPs into liver cancer cells, increasing the accumulation of *N*-14NCTDA. In vivo studies showed that SP94-LPs could effectively target to tumor sites and reduce the toxicity by avoiding the direct contact between *N*-14NCTDA and nontumor sites [122].

### 5.3. Lipid Microspheres

Lipid microspheres (LMs), also known as lipid emulsions, are oil droplets composed of soybean oil and lecithin, which have been used as a carrier for delivering lipophilic drugs due to their stable physical properties, good biocompatibility, and biodegradability. LMs can encapsulate drugs on the interface surface of the oil and aqueous phases to improve the solubility and stability of drugs, reduce the pain and irritation of intravenous drug injection, and enhance the therapeutic effect [29,123,124]. Lin et al. [125] prepared a NCTD venous lipid microspheres (NLM) to evaluate their pharmacokinetics, biodistribution, antitumor efficacy, and safety. Compared with the commercial product disodium norcantharidate injection (NI), NLM not only had similar pharmacokinetic characteristics, tissue distribution, and antitumor activity, but also significantly reduced cardiotoxicity and nephrotoxicity by avoiding the direct contact between NCTD and body fluids, thereby ensuring the effectiveness and safety of NCTD treatment. Ma et al. [126] prepared an NCTD–phospholipid complex (NPC) and encapsulated it in LMs to enhance the lipophilicity of NCTD and improve stability during the sterilization process. Tissue distribution studies indicated that NPCLM could improve tumor-targeting ability and reduce nephrotoxicity, which was associated with significantly increased lipophilicity of NPC. Pan et al. [127] demonstrated that lipid pellets containing C14 alkyl chain NCTD derivatives (*N*-14NCTDALMs) could not only improve stability and reduce the toxicity, but also prolong the biological half-life compared with NCTD and NCTD-LMs.

These studies show that NCTD lipid microspheres may be a promising strategy for increasing the stability and encapsulation rate of drugs, improving the bioavailability, enhancing the antitumor effect, and reducing the toxicity.

### 5.4. Polymer Micelles

Polymer micelles are a thermodynamically stable colloidal solution formed by the self-assembly of synthetic amphiphilic block copolymers in water. As drug carriers, polymer micelles can enhance the solubility and bioavailability of drugs, and the design of the hydrophilic shell structure can protect the drug from nonspecific absorption and prolong the retention time in the body, thereby enhancing the therapeutic effect [128]. Jiang et al. [129] prepared a Galactosamine-hyaluronic acid-Vitamin E succinate (Gal-HA-VES) amphiphilic polymer, which could self-assemble into multifunctional NCTD/Gal-HA-VES micelles to improve the therapeutic effect of liver cancer. Compared with free NCTD, NCTD/Gal-HA-VES micelles could enhance the antitumor activity and reduce the toxicity by enhancing liver targeting and increasing the accumulation of NCTD in tumor tissues. Zhang et al. [130] prepared arabinogalactan (AG)-modified N-(4-methimidazole)-hydroxyethyl (chitosan (MHC) polymer micelles, which could accurately deliver NCTD to the tumor site to improve antitumor activity. The polymer micelles could be used as an active liver-targeting carrier and improve the therapeutic effect. Compared with free NCTD, the polymer micelles of NCTD could increase intracellular drug accumulation by increasing cellular uptake and sustained the release of NCTD, thereby increasing HepG2 cell apoptosis and inhibiting cell invasion.

These results indicated that polymer micelles could specifically deliver NCTD to liver cancer tissues, which ensures the effectiveness and safety of tumor treatment. Therefore, polymer micelle is a promising drug delivery carrier for NCTD.

### 5.5. Thermosensitive Gel

Thermosensitive gel refers to a preparation that can quickly change from liquid to semi-solid gel with the change of temperature after being administrated in a liquid state, which is used as a chemotherapeutic agent delivery system due to its ability to increase drug accumulation at tumor sites, control drug release, and improve drug selectivity and antitumor activity in vivo [131,132]. Li et al. [133] developed NCTD-loaded metal–organic frame IRMOF-3, coated with a temperature-sensitive gel (NCTD-IRMOF-3-gel), to improve the antitumor activity of NCTD. Compared with free NCTD, NCTD-IRMOF-3-Gel could achieve sustained release of NCTD and prolong the circulation time in vivo. Furthermore, NCTD-IRMOF-3-Gel could block both S and G2/M phases of the Hepa1–6 cell cycle, significantly inhibit cell proliferation, and induce apoptosis. Gao B et al. [134] prepared NCTD nanoparticles (NCTD-NPs) encapsulated with doxorubicin (Dox) in a multi-F127 (PF127)-based hydrogel to construct a dual drug-loaded hydrogel system, which had good heat sensitivity, double sustained-release effect, and significant antitumor activity. Compared with free NCTD, this hydrogel system could enhance cytotoxicity to HepG2 cells and significantly inhibit cell proliferation by controlling the sustained release of NCTD and increasing cellular uptake. Meanwhile, this hydrogel system could also prolong survival of H22 tumor-bearing mice and reduce systemic toxicity. Xiao et al. [135] constructed an injectable thermosensitive N/O/hydrogel delivery system with slow-release capabilities to co-deliver NCTD and oxaliplatin (L-OHP) for treating cancerous ascites (Figure 12). Compared with free NCTD, the N/O/hydrogel system showed sustained drug release, good biocompatibility, and strong pro-apoptotic ability in vitro. In vivo studies showed that the N/O/hydrogel system could reduce the formation of ascites and tumor nodules, prolong the survival of tumor-bearing mice, and reduce systemic toxicity. Therefore, the N/O/hydrogel system is a promising platform for treating malignant ascites. Table 1 represents an overview of the nanodrug delivery strategies of NCTD.
Figure 12Schematic diagram on the synthesis of hydrogel drug delivery systems (N/O/hydrogel) and their treatment of tumors. Adapted from Ref. [135].
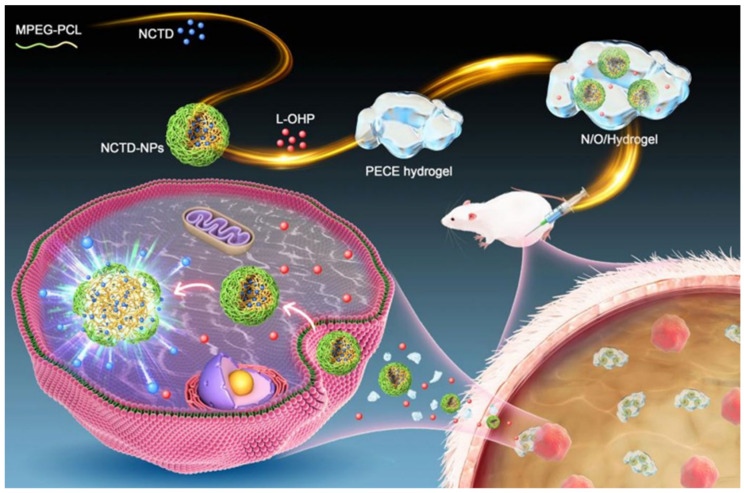



## 6. Conclusions

Compared with CTD, NCTD is a promising chemotherapeutic agent and has stronger biological activity and lower toxicity. However, it is limited in clinical application by poor solubility, low bioavailability, and nonspecific toxicity to normal cells. In order to overcome the limitations of NCTD, many strategies have been studied to improve the solubility and bioavailability of NCTD and reduce the toxicity. Firstly, optimizing the chemical structure of NCTD is a basic strategy to improve NCTD solubility based on the structure–activity relationship. Some NCTD derivatives, which were modified at the dicarboxylate anhydride structure or the C5/C6 position of NCTD, exhibited much stronger antitumor activity. However, the antitumor activity of the NCTD derivatives containing the chromone-linked pyrazole or -linked isoxazole moiety need to be further studied. Secondly, NCTD could combine with chitosan and its derivatives as well as small-molecule anticancer drugs to form prodrugs to improve the antitumor effects and reduce the toxicity. Therefore, the prodrug strategy of NCTD may be a promising strategy for reducing the toxicity and improving antitumor activity of NCTD. Thirdly, through solubility and bioavailability enhancement, toxicity reduction of NCTD could be achieved with the assistance of the microsphere dispersion system and NDDS. NDDS, especially active targeting NDDS, is also an ideal and promising method. The toxicity of NCTD, such as hepatotoxicity, nephrotoxicity, cardiotoxicity, and urinary system toxicity, as well as irritation to the injection site, are attenuated by NDDS. Furthermore, a new strategy for increasing in vivo efficacy and reducing systemic toxicity of NCTD is to encapsulate several components in NDDS, which would exert the synergistic therapeutic effect of multiple components [136]. Although these strategies have achieved some results in improving the solubility and bioavailability of NCTD and enhancing therapeutic efficacy, there are still some problems to be solved. For example, the action mechanism of some synthetic derivatives of NCTD is not clear, and the problems of poor stability of microspheres and liposomes have not been solved. Moreover, the biological distribution of NDDS and the long-term toxicity to normal tissues still need further investigation.

In conclusion, according to current studies, chemical structure change strategies have great potential in improving the solubility and antitumor activity of NCTD, while drug formulations have promising results in improving the solubility, bioavailability, and safety. Consequently, the future research direction of NCTD will focus on the synthesis of new derivatives of NCTD and the preparation of NDDS. Although faced with many difficulties, it is believed that with the continuous development of drug synthesis technology and nanotechnology, NCTD will show promising prospects in antitumor therapy.

## Figures and Tables

**Figure 2 molecules-27-07740-f002:**
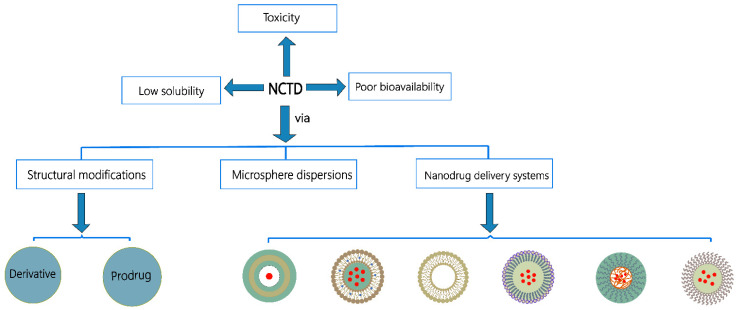
Strategies to improve solubility and bioavailability of NCTD.

**Figure 3 molecules-27-07740-f003:**
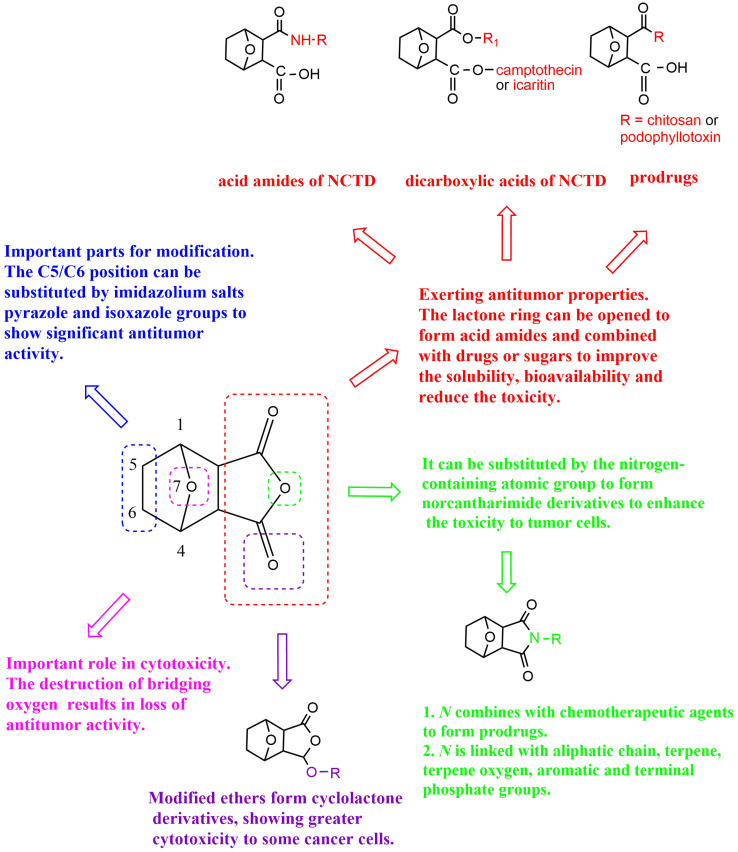
Structure–activity relationship and modification strategies of NCTD.

**Table 1 molecules-27-07740-t001:** Nanodrug delivery systems of NCTD.

Carrier Types	Materials	Experimental Subject	Properties	Ref.
Nanoparticles	PVP K30-coated NCTD chitosan	SD rats, ICR mice	Prolonged half-life and improved tissue distribution in mice.	[28]
	NCTD-loaded GC	SMMC-7721 and HepG2 cells, H22 mice	Enhanced toxicity to liver cancer cells, reduced the toxicity, and improved the intestinal absorption of NCTD.	[88,89]
	*N*-Trimethyl chitosan-encapsulated Lac-NCTD	HepG2 cells, H22 nude mice	Showed high liver targeting and strong antitumor effects in nude mice.	[90]
	NCTD-loaded mPEG-PCL-PEI-GA	HepG2 cells, tumor-bearing mice	Showed higher cytotoxicity and liver targeting, inhibited tumor growth, and prolonged the survival of tumor-bearing mice.	[93]
	NCTD-loaded RGD-LPH	TNBC cells, nude mouse	Inhibited TNBC tumor growth and metastasis.	[94]
	Strontium/Chitosan/Hydroxyapatite/NCTD Composite	MG-63 and MC3T3-E1 cells	Promoted osteogenesis and inhibited the proliferation of bone tumor cells.	[95]
	DMCA-Zn1 and DMCA-Zn2 nanoparticles	HepG2, Hep3B, and L927 cells	Inhibited cancer cell growth and proliferation and reduced the toxicity.	[26]
	NCTD-loaded modified lipid nanoparticles	HepG2 cells, SD rats	Inhibited HepG2 cell proliferation and induced apoptosis.	[96]
	DM-NCTD loaded in CHMSN and ABT-737 in lipid bilayer	H22 tumor-bearing mice	Showed stronger antitumor activity and reduced the toxicity in vivo.	[100]
	NCTD/Tet dual drug-loaded lipid nanoparticles	LO2, HepG2, HepG2/Adr and MCF-7cells	Showed strong antitumor effect by reversing the multidrug resistance and reduced the toxicity.	[101]
	NCTD/PTX-loaded core-shell lipid nanoparticles modified with APRPG	HepG2 cells, tumor-bearing mice	Remarkably inhibited the proliferation and migration of HCC cells.	[102]
Liposomes	NCTD liposomes modified with a novel mouse anti-human CD19 monoclonal antibody 2E8	BALB/c mice, Nalm-6 and Molt-3 cells	Specifically killed Nalm-6 cells and reduced the toxicity and was an effective method for treating B lineage hematopoietic malignancies.	[111]
		Nalm-6, Raji, Molt-3, and K562 cells	Specifically targeted B-LSCs and induced apoptosis.	[112]
	NCTD liposomes modified with a novel mouse anti-human CD19 monoclonal antibody 2E8b	HAL-01 and Molt-3 cells	Had the potential to specifically kill the B-LSCs and reduced the toxicity.	[113]
	NCTD-loaded SG-NCTD-LIP	HepG2 cells	Increased cytotoxicity to HepG2 cells.	[114]
	NCTD liposome–emulsion hybrid	H22 cells, SD rats, and H22 heterotopic or H22/Luc orthotopic tumor-bearing C57BL/6 mice	Controlled drug release, enhanced tumor-targeted accumulation, and reduced cardiotoxicity and nephrotoxicity.	[115]
	PH-sensitive liposomes loaded with Lac-NCTD phospholipid complex	HepG2 cells, H22 tumor-bearing mice	Showed better membrane permeability, higher capture rate, and stronger tumor-suppressive effects.	[118]
	DM-NCTD-loaded PEG/FA-PEG-liposomes	Kunming mice	Prolonged the circulation time of DM-NCTD in the blood and improved the bioavailability.	[120]
	Asialoglycoprotein receptor-targeted, galactosylated liposomes loaded with *N*-14NCTDA	HepG2 cells, SD rats, mice	Improved bioavailability and anticancer activity, reduced hepatorenal toxicity.	[121]
	*N*-14NCTDA-loaded liposomes modified with SP94	HepG2 cells, H22 tumor-bearing mice	Increased the liver-targeted accumulation of drug, improved the efficacy, and reduced the toxicity.	[122]
Lipid Microspheres	NCTD-loaded lipid microspheres	Male and female rats	Reduced cardiotoxicity and nephrotoxicity by avoiding the direct contact between NCTD and body fluids.	[125]
	NCTD–phospholipid complex	Kunming strain mice	Improved tumor-targeting ability and reduced nephrotoxicity.	[126]
	*N*-14NCTD-loaded lipid microspheres	SD rats	Improved stability, prolonged the half-life, and reduced the toxicity.	[127]
Polymer Micelles	NCTD-loaded self-assembled micelles	MCF-7, MCF-7/Adr and HepG2 cells, BALB/c nude mice	Showed high cytotoxicity in vitro, high liver targeting, strong antitumor effects, and low toxicity in vivo.	[129]
	arabinogalactan (AG)-modified *N*- (4-methimidazole) -hydroxyethyl (chitosan (MHC)	Male athymic nude mice	Increased the targeting of active liver drugs, improved the antitumor efficacy, and reduced the toxicity.	[130]
Thermosensitive Gel	NCTD-loaded metal–organic framework IRMOF-3 coated with a temperature-sensitive gel	Hepa1–6 cells	Showed sustained-release effects and inhibited the proliferation and induced apoptosis of Hepa1–6 cells.	[133]
	A thermosensitive hydrogel system co-encapsulated NCTD-NPs and Dox	H22 and HepG2 cells, Kunming female mice	Had significant anti-proliferative activity to HepG2 cells, inhibited tumor growth, and prolonged survival in H22 tumor mice.	[134]
	A thermosensitive N/O/hydrogel delivery system co-encapsulated NCTD-NPs and L-OHP	H22 and Huh7 cells, Kunming white rat	Showed stronger pro-apoptotic ability, prolonged the survival of tumor-bearing mice, and reduced systemic toxicity.	[135]

ICR, Institute of Cancer Research; SMMC-772, human hepatoblastoma cells; TNBC, triple negative breast cancer cells; MG-63, osteosarcoma cells; MC3T3-E1, osteogenic cells; L927, fibroblasts cells; LO2, human liver cells; HepG2/Adr, human drug-resistant liver cancer cells; K562 and HAL-01, human leukemia cells; Hepa1-6, mouse hepatoma cells; H22, murine hepatoma cells; Huh7, human hepatocellular carcinoma cells.

## Data Availability

Not applicable.

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
