# Peer review of "Strategies for Solubility and Bioavailability Enhancement and Toxicity Reduction of Norcantharidin"

_molecules, 2022, doi:10.3390/molecules27227740_

Round 1

Reviewer 1 Report

I would like to recommend the publication of this work after revision. However, the authors should solve the following issues.

1-All the abbreviations should be explained when used the first time in the manuscript. In addition, if you can avoid any of the abbreviations, it is preferred to write only full text.

2- The authors have written a meticulous review worth publishing. I would like to suggest the authors to add the references in 5.1. nanoparticles section  and mention about the followed published works:

Adlravan E, Nejati K, Karimi MA, Mousazadeh H, Abbasi A, Dadashpour M. Potential activity of free and PLGA/PEG nanoencapsulated nasturtium officinale extract in inducing cytotoxicity and apoptosis in human lung carcinoma A549 cells. Journal of Drug Delivery Science and Technology. 2021 Feb 1;61:102256.

-Samadzadeh S, Mousazadeh H, Ghareghomi S, Dadashpour M, Babazadeh M, Zarghami N. In vitro anticancer efficacy of Metformin-loaded PLGA nanofibers towards the post-surgical therapy of lung cancer. Journal of Drug Delivery Science and Technology. 2021 Feb 1;61:102318.

Author Response

Point 1: All the abbreviations should be explained when used the first time in the manuscript. In addition, if you can avoid any of the abbreviations, it is preferred to write only full text.

Response 1: According to the reviewer’s comments, we have explained all the abbreviations in the manuscript. For example, cancer cell line types were changed to the full text and the full text "protein phosphatase 1" of the abbreviation PP1 is added (line 153-154).

Point 2: The authors have written a meticulous review worth publishing. I would like to suggest the authors to add the references in 5.1. nanoparticles section  and mention about the followed published works:

Adlravan E, Nejati K, Karimi MA, Mousazadeh H, Abbasi A, Dadashpour M. Potential activity of free and PLGA/PEG nanoencapsulated nasturtium officinale extract in inducing cytotoxicity and apoptosis in human lung carcinoma A549 cells. Journal of Drug Delivery Science and Technology. 2021 Feb 1;61:102256.

-Samadzadeh S, Mousazadeh H, Ghareghomi S, Dadashpour M, Babazadeh M, Zarghami N. In vitro anticancer efficacy of Metformin-loaded PLGA nanofibers towards the post-surgical therapy of lung cancer. Journal of Drug Delivery Science and Technology. 2021 Feb

Response 2: We have added the above two papers (line 295, [80] and line 305, [87]). 

Reviewer 2 Report

The manuscript “Strategies for Solubility and Bioavailability Enhancement, Toxicity Reduction of Norcantharidin" is is a thorough overview of the chemical properties of Cantharidin and norcantharidin. The paper presents the main methods of deprivation of toxic molecules of norcantharidin. The data on the biological activity, in particular, the anticancer activity of the substances in question, are presented. The work makes a good impression, is written in a clear language and, in general, does not need edits. It is worth noting that the article is declared as an Article type, although it is a Review.

I think, this manuscript can be published in the Molecules in present form.

Author Response

The manuscript “Strategies for Solubility and Bioavailability Enhancement, Toxicity Reduction of Norcantharidin” is a thorough overview of the chemical properties of Cantharidin and norcantharidin. The paper presents the main methods of deprivation of toxic molecules of norcantharidin. The data on the biological activity, in particular, the anticancer activity of the substances in question, are presented. The work makes a good impression, is written in a clear language and, in general, does not need edits. It is worth noting that the article is declared as an Article type, although it is a Review.

Response:It is appreciated for your attention. This paper is a Review type and is indicated in the paper (line 1).

Reviewer 3 Report

This review manuscript discusses the research progress to increase the solubility and bioavailability,reduce the toxicity of norcantharidin, a natural product. The topic is interesting, and the manuscript is reasonably organized. I thus recommend its publication after addressing the following issues.

Point 1: For a more complete presentation of the material, it is required to bring the quantitative characteristics of norcantharidin, including biological half-life, solubility, etc.

Point 2: line 405-407: “Some derivatives of NCTD exhibited much stronger antitumor activity after structural modification, but the antitumor activity of other NCTD derivatives need to be further studied”. The derivatives of NCTD should be described in detail.

Point 3: line 407-408: “the prodrug of NCTD combined with small molecule anticancer drugs may be a promising strategy for reducing the toxicity and improving antitumor activity of NCTD”. What kind of small molecule anticancer drugs and how they combine with NCTD need to be further explained.

Point 4: The latest following important references should be added:

(1) Dong W, Wang X, Qian S, Wang Y, Zhao C. Regio-selective synthesis and activity research on 7-icaritin norcantharidin conjugates [published online ahead of print, 2022 Sep 12]. Nat Prod Res. 2022;1-9. doi:10.1080/14786419.2022.2121828

(2) Xie MH, Fu ZL, Hua AL, et al. A new core-shell-type nanoparticle loaded with paclitaxel/norcantharidin and modified with APRPG enhances anti-tumor effects in hepatocellular carcinoma. Front Oncol. 2022;12:932156. Published 2022 Sep 14. doi:10.3389/fonc.2022.932156

Author Response

Point 1: For a more complete presentation of the material, it is required to bring the quantitative characteristics of norcantharidin, including biological half-life, solubility, etc.

Response 1: We have added the specific quantitative data of biological half-life and solubility of NCTD according to the reviewer’s comments (line 39-40).

Point 2: line 405-407: “Some derivatives of NCTD exhibited much stronger antitumor activity after structural modification, but the antitumor activity of other NCTD derivatives need to be further studied”. The derivatives of NCTD should be described in detail.

Response 2: It is appreciated for your attention. According to your comments, we added the related content, “Some NCTD derivatives, which were modified at the dicarboxylate anhydride structure or the C5/C6 position of NCTD, exhibited much stronger antitumor activity. However, the antitumor activity of the NCTD derivatives containing chromone linked pyrazole or linked isoxazole moiety need to be further studied.” (line 526-529).

Point 3: 

line 407-408: “the prodrug of NCTD combined with small molecule anticancer drugs may be a promising strategy for reducing the toxicity and improving antitumor activity of NCTD”. What kind of small molecule anticancer drugs and how they combine with NCTD need to be further explained.

Response 3: According to the reviewer’s comments, we added the related content, “NCTD could combine with chitosan and its derivatives as well as small molecule anticancer drugs to form prodrugs to improve the antitumor effects and reduce the toxicity.” (line 530-531).

Point 4: The latest following important references should be added:

(1) Dong W, Wang X, Qian S, Wang Y, Zhao C. Regio-selective synthesis and activity research on 7-icaritin norcantharidin conjugates [published online ahead of print, 2022 Sep 12]. Nat Prod Res. 2022;1-9. doi:10.1080/14786419.2022.2121828

(2) Xie MH, Fu ZL, Hua AL, et al. A new core-shell-type nanoparticle loaded with paclitaxel/norcantharidin and modified with APRPG enhances anti-tumor effects in hepatocellular carcinoma. Front Oncol. 2022;12:932156. Published 2022 Sep 14. doi:10.3389/fonc.2022.932156.

Response 4: We are very sorry for our negligence of the two important papers and have added them in 3.1.1.4.dicarboxylic acid derivative section (line 185, [53]) and 5.1. nanoparticles section (line 351, [102]). In addition, we have added “icaritin” in dicarboxylic acid derivative section of NCTD in Figure 3 and the structure of NCTD derivatives from reference [52] in Figure 7 (derivative 6). The reference [102] has been added in Table 1.